# Modular assembly of proteins on nanoparticles

Wenwei Ma[1], Angela Saccardo[1], Danilo Roccatano [1], Dorothy Aboagye-Mensah[1], Mohammad Alkaseem[1], Matthew Jewkes[1], Francesca Di Nezza[2], Mark Baron[1], Mikhail Soloviev [3] & Enrico Ferrari [1]

Generally, the high diversity of protein properties necessitates the development of unique nanoparticle bio-conjugation methods, optimized for each different protein. Here we describe a universal bio-conjugation approach which makes use of a new recombinant fusion protein combining two distinct domains. The N-terminal part is Glutathione S-Transferase (GST) from *Schistosoma japonicum*, for which we identify and characterize the remarkable ability to bind gold nanoparticles (GNPs) by forming gold–sulfur bonds (Au–S). The C-terminal part of this multi-domain construct is the SpyCatcher from *Streptococcus pyogenes*, which provides the ability to capture recombinant proteins encoding a SpyTag. Here we show that Spy-Catcher can be immobilized covalently on GNPs through GST without the loss of its full functionality. We then show that GST-SpyCatcher activated particles are able to covalently bind a SpyTag modified protein by simple mixing, through the spontaneous formation of an unusual isopeptide bond.

[1] College of Science, University of Lincoln, Brayford Pool, Lincoln LN6 7TS, UK. [2] Department of Bioscience and Territory, University of Molise, Contrada Fonte Lappone, 86090 Pesche, Italy. [3] School of Biological Sciences, Royal Holloway University of London, Egham Hill, Egham TW20 0EX, UK. Correspondence and requests for materials should be addressed to E.F. (email: eferrari@lincoln.ac.uk)

Proteins possess a broad range of functional properties, including catalysis, cellular signaling, molecular recognition, and ligand binding. They are involved in virtually all biological processes with an unmatched range of physical and chemical properties, exceeding that of other biopolymers. Combining proteins with nanoparticles yields potentially new functional materials with a vast range of properties and applications. However, proteins bound to nanoparticles are not necessarily in their native environment and it is possible that conjugation can somehow compromise their functionality. For this reason, protein-nanoparticle bio-conjugation increasingly focuses on optimizing the orientation, accessibility and bioactivity of the conjugated molecule[1–4]. This is especially important in nanomedicine, as nanoparticles intended for drug delivery often fail to fully exploit their clinical potential due to limitations linked to their targeting ability and subsequent bio-distribution issues[5]. The goal of making nano-therapeutics that better retain the biological activity of bound biomolecules stimulated extensive studies of the interaction between nanoparticles and body fluids and the discovery of biological barriers that determine the fate of a nanoparticle within a living organism[6,7]. In drug delivery, the first biological barrier that nano-therapeutics face is given by the sequestration by phagocytic cells, which happens as a consequence of opsonization. This involves the adsorption of plasma proteins onto the surface of nanoparticles, the so-called protein corona[8]. It is widely accepted that the protein corona undergoes a dynamic exchange of molecules bound to the surface that depends on the exact proteins present in the specific body fluid, cellular compartment or environment[9–11]. Eventually, a stable "hard" corona forms on nanoparticles and this affects their uptake[12], cell association[13–15] and toxicity[16–18].

There exist a plethora of methods suitable for studying protein-nanoparticle interactions[19–22]. Expanding this knowledge could lead to the synthesis of designer recombinant proteins to facilitate the production of functionalized nanoparticles by forming a stable, artificial corona that modulates nanoparticle properties. This could be used to replace existing methods, for example based on pegylation, streptavidin-biotin interaction or tailored surface chemistry[23,24], which require the chemical modification of either the protein, the particle or both. A biotechnological, rather than purely chemical approach to bio-conjugation and particle surface modification could be advantageous when handling biomolecules intended for nanomedicine, especially in the case of protein therapeutics produced recombinantly.

Gold nanoparticles (GNPs) are of great importance in nanomedicine and have been used in nanoparticle-based biosensors[25,26]. Their interaction with proteins has been previously characterized using a combination of biochemical[27], biophysical[28] and computational approaches[29], therefore they represent an ideal model material for developing new decoration strategies. The ability of the engineered protein SpyCatcher to interact specifically with a SpyTag peptide has been used as a convenient protein–protein conjugation method[30]. The SpyCatcher/SpyTag pair has been shown to provide the ability to self-assemble and

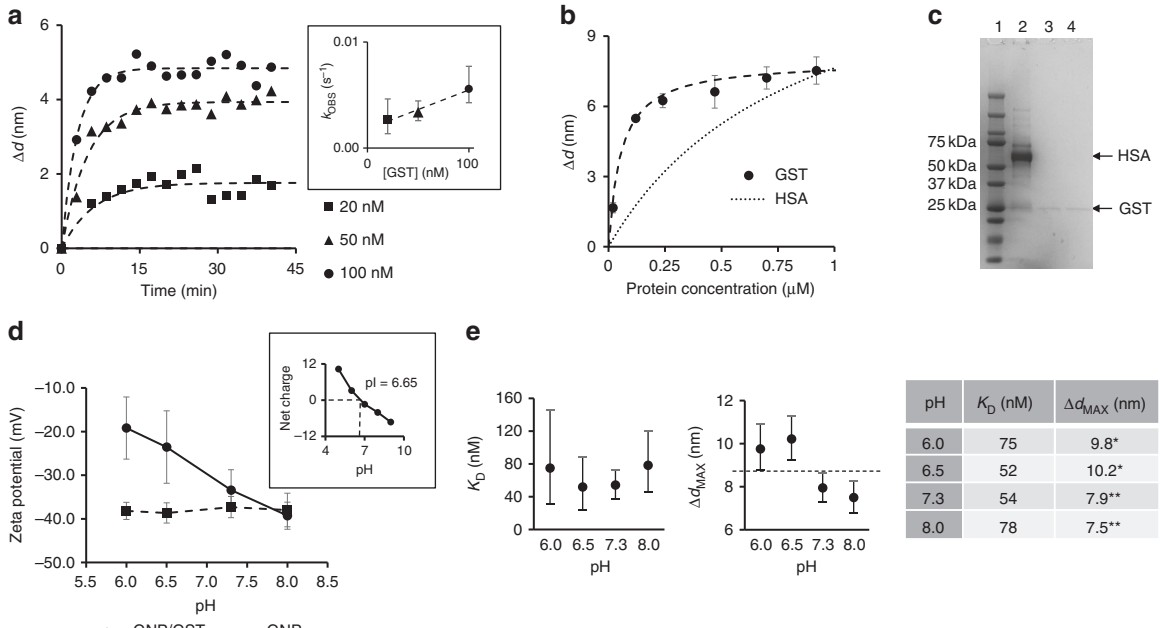

**Fig. 1** Binding of GST to GNPs. **a** GST binding measured using time-resolved DLS and fitted to a pseudo-first order association kinetic model (dashed lines) was used to determine the observed association rate $k_{OBS}$ at the GST concentrations indicated. The inset shows the values of $k_{OBS}$ and their 95% confidence interval (vertical bars) determined from each binding curve as described in Methods section. The dashed line in the inset represents the fit to the equation $k_{OBS} = k_{ON} [GST] + k_{OFF}$, from which the on- and off-rates of GST/GNPs binding were determined. **b** Equilibrium binding data showing that GST has higher affinity to GNPs than HSA. $\Delta d$ values for GST were averaged over three measurements and the error bars represent standard deviations. The dashed line is the best fit of GST data to the binding model used, whereas the dotted line is a representation of an ideal binding curve of HSA binding to GNPs, based on a previously reported $K_D$ and a $\Delta d_{MAX}$ of 14 nm. The latter was used assuming a complete corona with a thickness that equals the hydrodynamic diameter of serum albumin (~7 nm). **c** SDS-PAGE of GNPs incubated in human serum (HS). Lane 1: protein marker (relevant molecular weights are shown on the left); lane 2: unprotected GNPs incubated in HS showing prominent binding of HSA; lane 3: GST-protected GNPs incubated in HS showing no binding of HSA; lane 4: control GST-GNPs which were processed like lane 3 but not incubated in HS. **d** Zeta potential of GST-coated GNPs (solid line) measured at different pH, showing a marked increase at pH lower than pI. Naked GNPs (dashed line) are highly negative at all pH. The data are averages of three measurements and the error bars represent standard deviations. The inset shows the predicted net charge and pI of GST calculated using APBS algorithm at pH 5–9. **e** 95% confidence intervals of GST/GNPs $K_D$ (left), $\Delta d_{MAX}$ (center) and tabled data (right) at different pH. The $\Delta d_{MAX}$ estimates (also listed in the table inset) are significantly different, as indicated by the dashed horizontal line

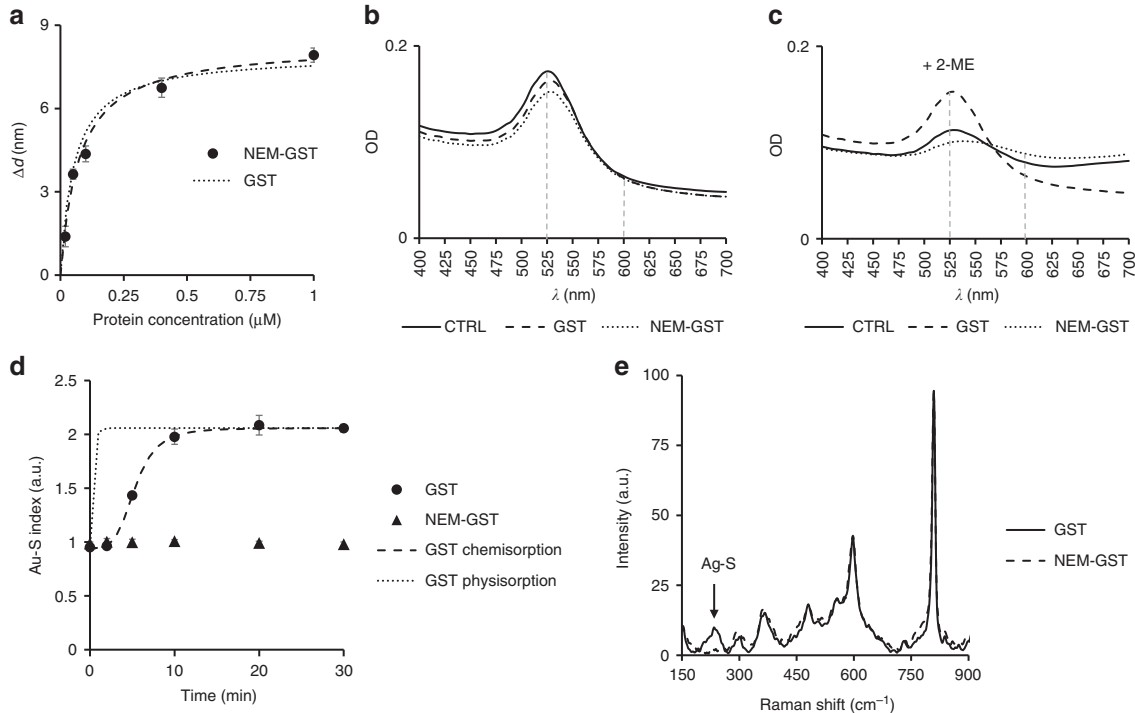

**Fig. 2** The role of GST thiols in the binding to GNPs. **a** Equilibrium binding data of NEM-GST fitted to the binding model (dashed line) show no significant difference with those of GST (dotted line, obtained from Fig. 1b; 95% confidence intervals for $K_D$ and $\Delta d_{MAX}$ estimates are represented in Supplementary Figure 3). **b** UV–vis spectra of naked GNPs (CTRL) compared to GNPs coated with GST or NEM-GST. Vertical dashed lines indicate the two wavelengths, 525 and 600 nm, that have been used for calculation of the Au-S index. **c** UV–vis spectra of the same colloids in **b** but after treatment with 2-ME. **d** Au–S stability index measured at different times shows an increased stability of GST/GNPs over time, whereas NEM-GST failed to protect from aggregation at any time point. GST chemisorption is represented as the fit of the Au–S index data to a sigmoid (dashed line). This is compared to the calculated physisorption trend (dotted line), obtained from Eq. 1 for the same GST concentration used in the chemisorption experiment. **e** SERS spectra of GST and NEM-GST adsorbed on silver colloid. The black arrow indicates the peak at 233 cm$^{-1}$ due to Ag–S mediated chemisorption. All data points with error bars in the figure represent average and standard deviation of three measurements

form a covalent isopeptide bond between the amino and carboxylic groups on the side chain of lysine and aspartate residues of the two distinct polypeptides. Interestingly, the formation of this unusual bond is self-catalyzed, making this one of the very few systems, together with SnoopCatcher/SnoopTag[31] and inteins[32] that allow protein–protein covalent conjugation without using chemical cross-linking. The ability to produce multi-protein mega-molecules from individual building blocks by using isopeptide bonds opened the way to applications well beyond protein biochemistry, for example in material science[33–35].

Here we describe a GST-SpyCatcher fusion protein that binds to GNPs due to the gold-binding ability of GST and allows the hierarchical assembly of an extra protein layer through the Spy-Catcher/SpyTag system. This modular approach to bioconjugation of recombinant proteins to nanoparticles does not require optimization of every specific protein-particle pair and provides universal platform for immobilizing functional proteins on gold nanoparticles.

## Results

**Binding properties of GST to GNPs.** We extensively characterized the interaction and adsorption of GST onto the surface of GNPs, as in our design this constitutes the primary interface between gold and subsequently attached proteins. GST binding was initially characterized by time-resolved dynamic light scattering (DLS), under the assumption that the increase of average particle diameter ($\Delta d$) was proportional to the amount of protein that binds to the particles[28] (see Supplementary Note 1 for a detailed explanation of the binding model we used). We

measured the time dependent increase of $\Delta d$ as a result of the formation of a protein corona (Fig. 1a). Incubation with different concentrations of GST determined a trend that was consistent with pseudo-first order association kinetics (see Eq. 1 in Methods section), and the observed rate ($k_{OBS}$) increased linearly with the concentration of GST (see the inset of Fig. 1a). Linear regression analysis of $k_{OBS}$ changes vs. GST concentration, revealed the on-rate ($k_{ON}$) and off-rate ($k_{OFF}$) which were $3.7 \times 10^4\,\mathrm{M}^{-1}\mathrm{s}^{-1}$ and $0.001789\,\mathrm{s}^{-1}$, respectively. Even at concentrations as low as 20 nM, the system reached an equilibrium within about 15 min, suggesting fast binding of GST to GNPs. For concentrations larger than 100 nM the equilibrium was too fast to determine a reliable association rate, as the equilibrium was reached within the very first data points.

To better determine the affinity of GST to GNPs across a broader range of concentrations, we measured $\Delta d$ at equilibrium and fitted the data to a saturation one-site binding model (see Eq. 2 in Methods section). The dissociation constant ($K_D$) and the plateau value for $\Delta d$ ($\Delta d_{MAX}$) obtained were 54 nM and 7.9 nm, respectively (Fig. 1b). Interestingly, the binding affinity of GST to gold was an order of magnitude higher than a previously reported value for human serum albumin (HSA), with an estimated $K_D$ of 833 nM[36]. Since HSA is the most abundant protein in serum, it is generally accepted that nanoparticles injected into the blood would be quickly covered by a protein corona, dominated by albumin and fibrinogen[37]. The remarkable affinity that GST has for gold suggests that it would be a good candidate to protect GNPs from rapid binding of serum proteins and subsequent opsonization. To test this, we incubated "naked" and GST-coated

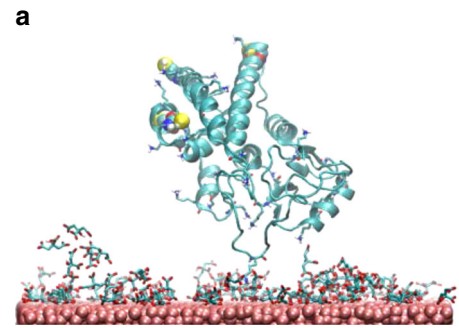

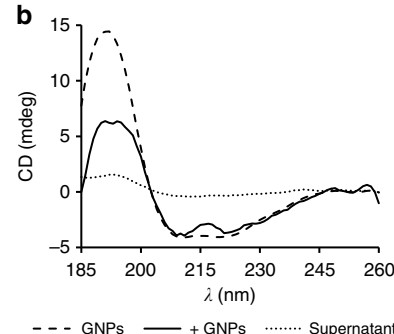

**Fig. 3** Rearrangement of GST bound to GNPs. **a** Representative conformation after 20 ns MD simulation of GST (PDB ID: 1UA5) bound to citrate-capped gold surface. GST has rotated compared to the original position (Supplementary Figure 4B) to orient lysine residues toward the citrate on the surface. Positively charged residues and cysteines are shown as stick and van-der-Waals representation, respectively. The complete simulation space of this and two other simulation events is reported in Supplementary Figure 4C-E. Supplementary Figure 4F also shows that there are no evident secondary structure changes within 20 ns of simulation. **b** SRCD spectra in the far UV region of GST in solution (dashed line) compared to GST bound to GNPs (solid line) measured at the same concentration. The latter was incubated with excess GST for 1 h and extensively washed before measurement to avoid the presence of protein in solution. After measurement GNPs were removed by centrifugation and a spectrum of the supernatant only (dotted line) was acquired and showed that little, if any, protein was present in solution, confirming that the spectrum of GST/GNPs was representative of the conformation of GST on gold surface. Similar measurements conducted on silver colloid are shown in Supplementary Figure 5

GNPs in human serum for 3 h and compared the pattern of bound protein by SDS-PAGE (Fig. 1c). Whereas unprotected GNPs show extensive binding of proteins, resembling the same pattern of human serum with prominent amount of HSA (Supplementary Figure 2), the particles previously protected with GST did not bind further proteins, providing evidence that GST can strongly influence the formation of a protein corona in serum.

We have assessed whether GST-coated GNPs were responsive to pH changes by measuring their zeta potential at different pHs (Fig. 1d). We found that naked GNPs had a highly negative potential due to citrate capping and thus were essentially not responsive to pH change. On the contrary, GST-coated GNPs showed a marked increase of potential at lower pH, similar to what was previously observed for GNPs exposed to plasma proteins[38]. This trend was also consistent with the predicted net charge of GST at different pHs, estimated using an Adaptive Poisson-Boltzmann Solver (APBS) algorithm[39], which gave a theoretical isoelectric point (pI) of 6.65 (inset of Fig. 1d). Positively charged GST at pH 6 and 6.5 compensated the highly negative potential of GNPs; whereas, the potential at pH above the GST pI was nearly unaffected by the presence of GST. To further assess how pH affects GST binding properties, we repeated the equilibrium binding measurement of Fig. 1b, which was taken at pH 7.3, for the same pH values for which we measured the zeta potential. We fitted the data using Eq. 2 (Methods section) and the results are reported in Fig. 1e.

$K_D$ values at different pH were found to be not significantly different, whereas $\Delta d_{MAX}$ values were found to belong to two significantly different groups, with the two values obtained at pH below pI being significantly higher than those above pI. According to the binding model used, higher values of $\Delta d_{MAX}$ indicate a larger average number of molecules bound per particle. The higher protein load at lower pH could be explained by a more dense protein packing on the GNP surface, likely facilitated by the net positive charge of GST at lower pH, which can partially compensate the negative charges due to citrate capping on the gold surface. As maximizing the protein load was not the scope of our design, we used pH 7.3 for all the subsequent experiments, as this is the pH at which the proteins we used were stably stored and it is also more representative of physiological conditions compared to pH 6.0 or 6.5.

**Binding mechanism of GST to GNPs**. Our initial findings indicated a stable interaction at the GST-GNP interface. We therefore assessed whether GST binds to GNPs by physisorption or chemisorption, with the latter known to provide more stable GNP bioconjugates. We studied the mechanism of binding with a specific focus on the thiol groups at the 4 cysteine residues of GST, as these have been reported to form covalent Au–S bonds in other proteins[40]. First, we compared the binding properties of GST and N-ethylmaleimide (NEM)-modified GST (NEM-GST). NEM was used to convert the sulfhydryl at cysteine residues into thioester groups, so the potential formation of Au–S bonds and their involvement in the binding mechanism to gold could be investigated. We found that $K_D$ and $\Delta d_{MAX}$ of NEM-GST were 73 nM and 8.3 nm, respectively. Interestingly, these are not significantly different compared to the values obtained for GST, suggesting that thiols are not essential for the initial adsorption of GST onto GNPs (Fig. 2a and Supplementary Figure 3).

We evaluated the stability of the colloids by measuring the absorbance near the surface plasmon resonance (SPR) peak at 525 nm. As expected, addition of either GST or NEM-GST did not destabilize the colloid, as evidenced by a clear SPR peak near 525 nm in all the preparations tested (Fig. 2b). However, the addition of 2-mercaptethanol (2-ME) resulted in a pronounced reduction of the SPR of naked GNPs and of those bound to NEM-GST; whereas, GST-coated GNPs were unaffected (Fig. 2c). This suggests that 2-ME can destabilize the colloid by binding to the gold surface and displacing NEM-GST. However, 2-ME could not displace GST with active thiol groups, most likely because they form a covalent bond with the gold surface. We defined an Au–S index by measuring the ratio between the absorbance near the SPR peak (525 nm) and the absorbance at a wavelength far from the peak (600 nm). Instability of the colloid in 2-ME would result in aggregation and decreased absorbance at the SPR peak, resulting in an Au–S index of ~1, as an aqueous buffer without a colloid suspended would have similar absorbance at 525 nm and 600 nm. Values larger than 1, instead, indicate presence of colloid highly stable in 2-ME, as observed in the case of GST. Here, the SPR peak was twice higher than the absorbance at 600 nm (Fig. 2c), giving an Au–S index of ~2. The assay wavelength was chosen to be sufficiently far away from typical SPR wavelengths of gold and silver nanoparticles and yet be compatible with a broad range of absorbance spectrometers including instruments used for turbidity assays.

We further evaluated the kinetics of GNPs stabilization using the 2-ME assay. GNPs were incubated with either GST or NEM-GST for increasing defined times prior to 2-ME treatment and the stability index was calculated for each time point (Fig. 2d). Whereas NEM-GST was unable to stabilize the colloid irrespective of the incubation time, GST was able to fully protect GNPs from aggregation, but only following an incubation time of ~10 min, as evidenced by the increase of the Au–S index from 1 to 2. This is a strong indication that, although the 4 thiols on GST do not contribute to the initial rapid adsorption on gold (physisorption), they bind and stabilize the GNPs in the longer term, presumably by formation of covalent bonds (chemisorption). We fitted the Au–S index time course data with a sigmoid model (see Eq. 3 in Methods section) and obtained a half-time of 5.3 min, which represents the time required for the Au–S stability index to reach half of the plateau value. This time is much longer than the half-time of 12 s calculated from the on-rate estimation of Fig. 1a, suggesting a multi-step process of binding of GST to GNPs, which relies on multiple mechanisms having very different kinetics. This observation is consistent with a three-steps model of adsorption previously described only qualitatively[41], which involves an initial reversible association step (presented in Fig. 1), a rearrangement/reorientation step on the gold surface, and a final cysteine-dependent "hardening" step, after which binding becomes irreversible. We then used surface enhanced Raman

scattering (SERS) on silver colloid-adsorbed protein to provide extra evidence of cysteine involvement in the mechanism of binding of GST to metal nanoparticles. Comparison of the Raman spectra of GST and NEM-GST adsorbed on citrate-capped silver nanoparticles (SNPs) revealed a Raman peak at 233 cm$^{-1}$ only in the GST sample (Fig. 2e). This peak was previously attributed to Ag–S stretching mode in the SERS of thiols on silver colloid[42] and also in the case of proteins adsorbed on SNPs[43,44]. These data further support the idea that the different interaction that GST and NEM-GST have with metal surfaces is likely due to thiols availability to form a covalent bond.

To better understand the rearrangement/reorientation step, we used molecular dynamics (MD) to model likely orientations of GST on a citrate capped gold surface. The results indicated that all the conformations obtained within the time scale of the simulations (20 ns) relied on lysine residues forming salt-bridges with citrate molecules (Fig. 3a and Supplementary Figure 4). This is consistent with our previous observation that positive charges on GST surface play a role in adsorption (Fig. 1d–e) and Au–S bond formation is not the initial driving force for adsorption (Fig. 2d).

To establish the nature of the subsequent GST rearrangement and the degree of conformational changes necessary for Au–S formation, beyond the time scale of a MD simulation, we compared the secondary structure of GST in solution with that of

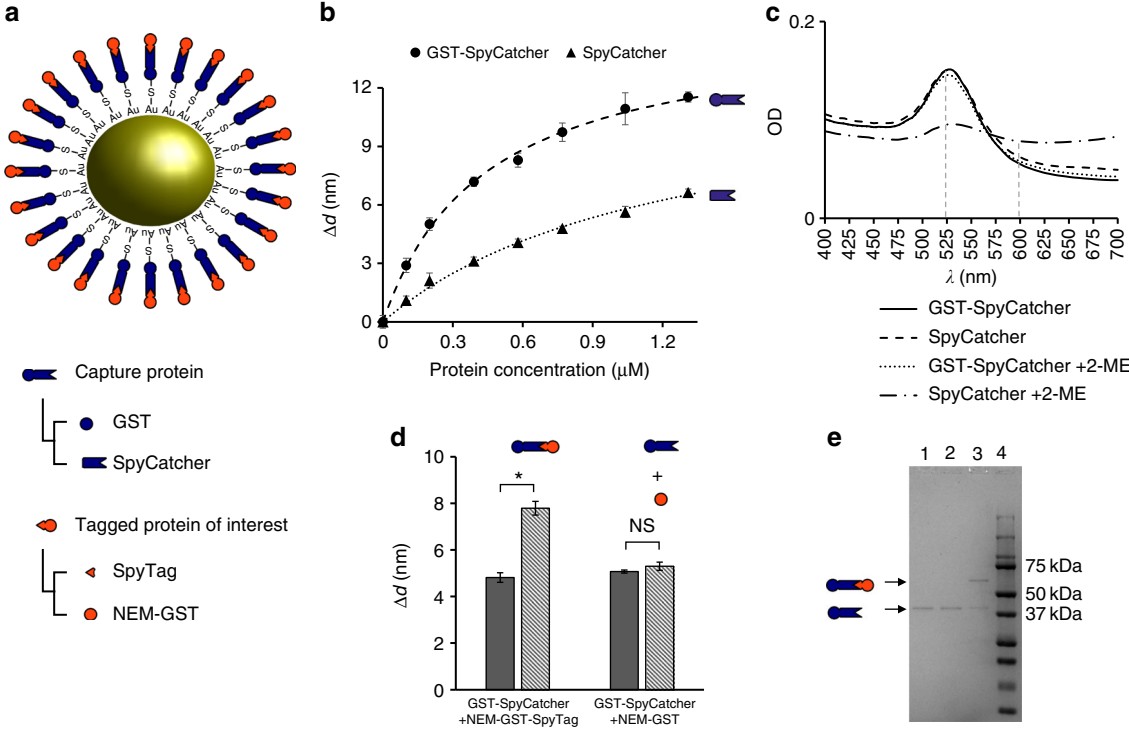

**Fig. 4** Hierarchical assembly of GST-SpyCatcher and SpyTag on GNPs. **a** Design of the conjugation strategy. Blue identifies the first layer of proteins, orange highlights the second layer. Figure is not to scale. **b** Equilibrium binding data of GST-SpyCatcher compared to SpyCatcher without GST. The fit to the binding model shows that GST fusion improves binding of SpyCatcher to GNPs. Data points and error bars represent average and standard deviation of three measurements respectively. **c** Comparison of UV–vis spectra of GNPs coated with GST-SpyCatcher or SpyCatcher before and after adding 2-ME. Whereas SpyCatcher/GNPs are destabilized by 2-ME, GST-SpyCatcher/GNPs are stable, suggesting formation of Au–S bonds. Vertical dashed lines indicate the two wavelengths previously used for calculations of the Au–S index and highlight a sharp decrease of the SPR peak at 525 nm in absence of GST. **d** Δ$d$ of GST-SpyCatcher/GNPs before (dark gray) and after (light gray) addition of either NEM-GST-SpyTag (left) or untagged NEM-GST (right). Tagging NEM-GST with SpyTag results in a significant increase of Δ$d$, suggesting that a second layer of protein extensively binds to GNPs. Δ$d$ are averaged over three measurements with error bars representing standard deviation. $t$-test results with a 95% confidence interval are indicated by asterisk (significant, $p = 0.0001$) and NS (non-significant, $p = 0.0991$). **e** SDS-PAGE of proteins bound to GNPs. Lane 1: GST-SpyCatcher/GNPs before any further incubation; lane 2: GST-SpyCatcher/GNPs incubated with untagged NEM-GST; lane 3: GST-SpyCatcher/GNPs incubated with NEM-GST-SpyTag; lane 4: protein marker (relevant molecular weights are reported on the right). A band corresponding to GST-SpyCatcher-NEM-GST-SpyTag appears in lane 3, suggesting formation of the isopeptide bond between the two layers of protein corona

GST-GNPs using synchrotron radiation circular dichroism (SRCD)[45] (Fig. 3b). We found that, whereas the overall secondary structure was well preserved, there were some changes in the spectrum in the presence of GNPs, especially a marked reduction of the SRCD value around 190 nm. This is consistent with an increased contribution of disordered/irregular conformation, which would reduce the positive band at about 190 nm associated to the α-helical conformation of GST in solution. We concluded that the rearrangement of GST on GNPs leads to partial denaturation of the GST secondary structure. A similar change in the structure of GST bound to silver colloid was detected, suggesting a similar interaction (Supplementary Figure 5). It is still debated to what degree proteins on nanoparticles do denature and what the actual impact on their function is. For example, it has been reported that albumin-specific antibodies can bind albumin adsorbed on GNPs, suggesting that denaturation is not dramatic and does not affect all of the native epitopes[10,45].

In our design, GST serves merely as the interface with gold and the preservation of its native transferase activity is not required. Therefore, we focused on the ability of GST to bind gold, which was observed before[46] but not extensively characterized. We concluded that GST binds to GNPs with high affinity in a predictable, oriented manner. GST forms a covalently attached protein corona on GNPs and enables the attachment of the next hierarchical level of proteins.

**Hierarchical and controlled assembly of a protein corona.** The nanoparticle-protein interface was further modified by adding SpyCatcher to the C-terminal of the GST. We set out to verify whether or not gold-immobilized SpyCatcher retains its spontaneous interaction with SpyTag and whether the fusion protein GST-SpyCatcher retains its ability to irreversibly bind GNPs. We therefore tagged a protein of interest, in this specific case NEM-GST, with a SpyTag peptide and probed the ability of NEM-GST-SpyTag to bind GST-SpyCatcher activated GNPs (Fig. 4a).

Initially, we measured $K_D$ and $\Delta d_{MAX}$ of GST-SpyCatcher in an equilibrium binding experiment, to assess whether the protein binds effectively to GNPs (Fig. 4b). We found a $K_D$ of 438 nM and a $\Delta d_{MAX}$ of 15.3 nm, both larger than GST alone. The higher dissociation constant might be attributed to the lower pI of GST-SpyCatcher (pI = 5.17) compared to GST (pI = 6.65), and therefore a lower propensity to bind negatively charged particles. The larger $\Delta d_{MAX}$ is consistent with the larger size of the protein (GST-SpyCatcher is 38.5 kDa compared to 28.6 kDa of GST). We also measured $K_D$ and $\Delta d_{MAX}$ of SpyCatcher not fused to GST and found values of 1.21 µM and 12.5 nm, respectively. Whereas $\Delta d_{MAX}$ was slightly, but not significantly smaller than GST-SpyCatcher, $K_D$ was significantly higher (see confidence intervals in Supplementary Figure 6). This suggests that GST has a positive role in affecting the affinity of fusion proteins to GNPs. We verified whether GST can mediate the chemisorption of GST-SpyCatcher by measuring the SPR peak in the presence of destabilizing 2-ME (Fig. 4c). The results show that, whereas SpyCatcher alone was displaced by 2-ME, the GST-SpyCatcher fusion was stable, suggesting formation of Au–S bonds at cysteine residues on GST (SpyCatcher does not contribute any cysteine), making GST-SpyCatcher an ideal protein to form the first layer of the hierarchical protein corona.

To prove the ability of GST-SpyCatcher to assemble a second layer of protein on GNPs, SpyTag was fused to GST and its binding to GNP-GST-SpyCatcher was studied. Cysteines of GST-SpyTag were inactivated using NEM (NEM-GST-SpyTag) to rule out any possible chemisorption that would make the interpretation of binding data more difficult. We exposed two preparations of GNP-GST-SpyCatcher to excess of NEM-GST-SpyTag or untagged NEM-GST and measured the resulting $\Delta d$ after 3 h incubation (Fig. 4d). Whereas NEM-GST did not significantly increase the overall size of the particles, NEM-GST-SpyTag did bind to the activated particles and increased the diameter by 3 nm. The proteins that were bound to the GNPs of Fig. 4d were loaded on SDS-PAGE to verify whether NEM-GST-SpyTag formed the expected isopeptide covalent bond with GST-SpyCatcher. A prominent band was found compatible with the expected molecular weight of a GST-SpyCatcher/NEM-GST-SpyTag complex which is 66.5 kDa (38,5 kDa from GST-SpyCatcher and 28 kDa from NEM-GST-SpyTag). We concluded that NEM-GST-SpyTag was able to bind to GST-SpyCatcher activated GNPs and form a second level of protein stably associated to GNPs.

This step-wise self-assembly method could be applied in principle to other gold nanomaterials (for example nanorods or gold nanoelectrodes) or even materials other than gold. For example we replicated the DLS experiment and SDS-PAGE of Fig. 4 (**d** and **e**, respectively) using citrate-capped silver nanoparticles (SNPs) in place of GNPs and we obtained very similar results, which are reported in Supplementary Figure 7. Activation of surfaces other than gold and silver could be potentially explored by fusion of SpyCatcher to established protein-material interfaces other than GST. For example peptides that are able to selectively bind to carbon nanotubes, polystyrene, silica and many other materials have been previously reported[23,47].

## Discussion

In summary, we have achieved covalent immobilization of proteins onto GNPs using a two-steps approach. The first step relies on GNPs modification mediated by GST through Au–S cross-linking. The second step relies on spontaneous covalent formation of a unique isopeptide bond between SpyCatcher and the SpyTag peptide. The latter can be easily added to any recombinant protein. The GNP-GST-SpyCatcher complex is stable and thus provides a universal platform for decorating GNPs, using a simple and easily reproducible protocol. We designed this robust conjugation strategy primarily for the bio-conjugation of recombinant proteins to nanoparticles or other surfaces, with applications in nanomedicine and biosensors in mind and a particular emphasis on preserving protein structure and function throughout the process. Several other protein-particle conjugation approaches have been extensively reviewed recently[3,23]. The most common covalent method applied to gold and silver nanoparticles makes use of Au–S or Ag–S bonds from cysteine residues, either native or deliberately introduced within a recombinant protein. Although this method benefits from the high stability and spontaneous nature of the interaction, the protein structure is often affected by mutagenesis and by strong interaction of the cysteines with the metal nanoparticle[3]. The close proximity and extensive physical contact of the recombinant protein with the nanoparticles often has detrimental effect on the structure and function of the immobilized protein[48]. Our approach takes advantage of the unique gold-binding properties of GST whilst also limiting the risk of structural interference with the protein of interest through oriented immobilization and by acting as a spacer to distance the protein from the gold surface. A common alternative to covalent protein-nanoparticle conjugation is represented by the biotin–avidin chemistry and the several derivatives and homologs, due to the unusually high affinity between the two components[3,23]. These systems are often the first choice when prototyping bioconjugates, as many commercial products are already available. Biotin–avidin interaction is almost ubiquitous in molecular diagnostics and research; however, the

simplicity of this strategy comes with limitations when using bioconjugates in nanomedicine, such as (i) heterogeneity of labeling due to the tetravalent nature of avidin, (ii) poor control on binding site of biotin during chemical cross-linking[3] and (iii) interference of endogenous biotin with conjugates in vivo, as the concentration of biotin (also known as vitamin H) is not negligible in serum and tissues[49]. Site-specific recombinant inclusion of a designer conjugation tag, such as SpyTag is an increasingly common choice for protein bioconjugation[35] and proved to successfully preserve structure and/or function of target proteins in several contexts, such as embedding enzymes in nanovesicles[50], displaying target antigens on virus-like particles[51], enclosure of cargo proteins in phage capsids[52], conjugation of affibodies to fluorescent proteins[53], protein labeling for super-resolution imaging[54], supramolecular assembly of multi-enzyme metabolic pathways[55], conjugation of a DNA polymerase to a nanopore[56], and protein hydrogels[33,34]. SpyCatcher protein has been successfully used in combination with quantum dots to make protein-nanomaterials conjugates[57,58]. In the latter works, SpyCatcher was modified by introducing two cysteine residues at the N-terminal specifically for conjugation purpose. We opted for a GST-fusion of the unmodified SpyCatcher polypeptide instead. This brought about a substantially increased gold-binding affinity of the fusion protein due to the presence of GST tag, as clearly indicated by equilibrium binding curves obtained for SpyCatcher and GST-SpyCatcher (Fig. 4b).

The ability of GST to adsorb effectively onto gold nanoparticles and to provide a stable anchor for SpyCatcher, makes development of complicated surface chemistries unnecessary. Citrate capped metal nanoparticles can easily be synthesized in the lab or manufactured at large scales. All conjugation experiments could be conducted in aqueous buffers at neutral pH. Expression and purification of GST-SpyCatcher in bacteria is standard and scalable procedure, suggesting that the overall strategy can be potentially used industrially. Importantly, as nanomaterials are likely to receive closer attention from pharmaceutical regulatory bodies[59], the use of metal nanoparticles that are already well-established in nanomedicine[60–62] and do not carry extra chemical modifications at the surface, makes this bio-conjugation method suitable for consideration in nanopharmaceutical projects.

## Methods

**Preparation of nanoparticles**. Citrate capped 40 nm GNPs were purchased from BBI Solutions at a concentration of 5 OD in water (corresponding to $4.5 \times 10^{11}$ particles ml$^{-1}$ according to the provider). For the SRCD experiment we used citrate capped 20 nm GNPs purchased from Sigma-Aldrich at a stock concentration of 1 OD in 0.1 mM PBS (corresponding to $6.5 \times 10^{11}$ particles ml$^{-1}$ according to the provider). Citrate capped 40 nm silver nanoparticles were purchased from Sigma-Aldrich at a concentration of 0.02 mg ml$^{-1}$ in aqueous buffer. For protein adsorption experiments, nanoparticles were washed by centrifugation, re-suspended in 10 mM HEPES, pH 7.3, 10 mM NaCl and their concentration measured and normalized at the desired OD using NanoDrop 2000 (Thermo Fisher Scientific). For the initial characterization of GST adsorption at different pH, phosphate buffers at the indicated pH were used in place of HEPES. Human serum used in nanoparticle binding experiments was from Sigma-Aldrich.

**Synthesis of proteins**. All proteins were synthesized recombinantly using pGEX-KG GST gene fusion system (Addgene) expressed in BL21(DE3)pLysS E.coli strain (Thermo Fisher Scientific). The resulting amino acid sequences of all the proteins used in this paper are reported in Supplementary Table 1 along with the relevant European Nucleotide Archive (ENA) accession numbers of each coding DNA sequence used. All recombinant proteins were purified from bacterial lysates by affinity chromatography using Glutathione Sepharose 4B resins (GE Healthcare). SpyCatcher was obtained by thrombin cleavage at a proteolytic site between GST and SpyCatcher and elution from the affinity resin. All proteins were further purified by size exclusion chromatography using Superdex 75 10/300 GL columns on an ÄKTA Pure chromatography system (GE Healthcare). Protein concentration was determined with BCA assay (Thermo Fisher Scientific). Inactivation of thiols of GST and GST-SpyTag was achieved by treating 30 μM protein with 1.2 mM NEM for 2 h at room temperature. Excess of NEM was removed by using Zeba spin desalting columns with a molecular weight cutoff of 7 kDa (Thermo Fisher

Scientific). The inactivation was verified by Ellman's reaction using 4 mg of 5,5′-Dithiobis(2-nitrobenzoic acid) (Sigma-Aldrich) dissolved in 1 mL of 0.1 M sodium phosphate buffer pH 8.0 and 1 mM EDTA. Ellman's reagent was mixed with protein in a 3:1 ratio for 10 min at room temperature, the absorbance was measured at 412 nm and compared with the values obtained from a standard curve made using L-cysteine. The inactivation was considered successful if the absorbance was suggesting negligible presence of active thiols.

**Dynamic light scattering and zeta potential measurements**. All measurements were performed at 20 °C using Zetasizer Nano ZS (Malvern), using disposable micro-cuvettes or folded capillary cells (zeta potential) and automatic optimization of parameters. In DLS measurements, the diameter ($d$) was defined by the Z-average particle size and the increment of diameter ($\Delta d$) was calculated by subtracting the Z-average of the naked nanoparticles to each measurement. Increments were measured in the same cuvette by increasing the protein concentration every 30 min of incubation (equilibrium data) or, for kinetic data, particle size was recorded continuously and immediately after adding the protein, except for the value at time 0, which was recorded before addition of protein.

**Surface plasmon resonance**. Absorption spectra were recorded using an Infinite M200 PRO plate reader (TECAN) from 200 μL suspensions of GNPs at a final concentration of 0.4 OD, incubated for a determined duration with the relevant protein, before and after addition of 2-ME at a final concentration of 0.2 mM.

**Surface enhanced Raman scattering**. Protein was adsorbed on citrate capped SNPs, concentrated and washed as described above. To improve SERS signal, protein-SNPs were mixed with hydroxylamine phosphate silver nanoparticles in aqueous buffer, prepared as previously described[63] in a volume of 500 μl in a glass cuvette (0.5 cm path length). Raman spectra were recorded using a LABRAM 300 (Horiba Jobin Yvon) with an excitation line of 532 nm, equipped with an Olympus microscope BX41. The laser output power used was 50 mW for 10 s of exposure time (2 accumulations).

**Molecular dynamics**. The structure of GST was obtained from the Protein Data Bank (PDB ID: 1UA5 [https://doi.org/10.2210/pdb1UA5/pdb]). The ligand glutathione and sulfate ions were removed while the crystallization water was kept. The crystal structure does not contain the C-terminal sequence which is part of the pGEX-KG expression vector (see Supplementary Table 1), therefore this part was initially modeled from the amino acid sequence using Modeller[64] which yielded a random coil. All the molecular dynamics simulations were performed using GROMACS (version 5)[65,66], VMD[67] was used for visualization. MD simulation conditions are detailed in Supplementary Methods. The initial structure was equilibrated in absence of gold surfaces for 100 ns and its conformation, inclusive of the extra amino acids from the expression vector, is reported in Supplementary Figure 4A. The surface of the nanoparticle was approximated to two opposite planar slabs of gold atoms each consisting of four 10 × 10 nm layers of gold atoms. Since the average diameter of the nanoparticles used in the experiment was ~40 nm the effect of their curvature is negligible for a protein of ~2.4 nm in size. The two parallel gold slabs formed the opposite faces on the longest side of a rectangular box of size 10 × 10 × 12 nm. The citrate coating of the surfaces was obtained using 384 citrate molecules initially located in the middle of the simulation box. The box was then filled with water and sodium counter-ions to neutralize the total charge of the system and 20 ns MD simulation allowed the two Au surface to be coated with ~100 citrate molecules. Water, citrate molecules and sodium ions were removed from the central region of the box. The protein conformation obtained after 100 ns of simulation in solution, comprising a 0.6 nm thick solvation shell, was placed in the center of the box and oriented with the longest principal axes parallel to the Au surface as shown in Supplementary Figure 4. The remaining empty space was refilled with water molecules and the total charge of the box was adjusted to neutrality and, finally, binding of GST to GNPs was simulated for 20 ns in three distinct simulations with different starting velocity. The compositions of all simulated systems are reported in Supplementary Table 2.

**Synchrotron radiation circular dichroism (SRCD)**. SRCD spectra were measured at B23 beamline for SRCD of Diamond Light Source (UK)[68] and processed using CDApps[69]. A cuvette cell (Starna) of 2 mm path-length was filled with high-OD nanoparticles re-suspended in 0.1 mM HEPES. The beamline's Zetasizer ZSP (Malvern) was used before SRCD measurements to verify presence of the expected protein corona after extensive washes. Spectra were averaged over three measurements with a 1 s integration time.

**Gel electrophoresis**. Nanoparticles at OD 5 were incubated with the relevant proteins, washed twice in 10 mM HEPES, pH 7.3, 10 mM, NaCl by centrifugation at 5000×g for 5 min and re-suspended in 15 μl buffer. Washed nanoparticles were mixed with 5 μl of 4× SDS-PAGE loading buffer (0.06 M Tris, pH 6.8, 10% glycerol, 2% SDS, 0.005% bromophenol blue, 5% 2-ME), heated at 95 °C for 5 min and left at 42 °C for 10 h. Proteins that were originally bound to the particles were fully recovered after removal of GNPs by centrifugation at 5000×g for 5 min and loaded

on a 12% pre-cast SDS-PAGE gel (Expedeon) along with a protein ladder (Precision Plus Protein Dual Color Standard, BioRad). The gel was stained for 6 h by InstantBlue (Expedeon) and destained in water for 10 h before imaging.

**Curve fitting and statistics**. All kinetics and equilibrium binding curves were fitted using a previously described nonlinear least-squares method[70]. The same method was also used to estimate the confidence interval of each parameter and the statistical significance of the differences between them. Kinetics $\Delta d$ data acquired at different times ($t$) were fitted to equation 1 to retrieve observed rate at a determined concentration ($k_{OBS}$) and maximum size increment observed at the same concentration ($\Delta d_{OBS}$).

$$\Delta d = \Delta d_{OBS}(1 - e^{-k_{OBS}t}). \tag{1}$$

Equilibrium binding data were fitted to equation 2 to estimate $\Delta d_{MAX}$ and $K_D$ from $\Delta d$ data obtained at different protein concentrations [P].

$$\Delta d = \Delta d_{MAX}\frac{[P]}{K_D + [P]}. \tag{2}$$

The 4-parameters sigmoid of equation 3 was used for the estimation of the half-time (c) of Au–S bond formation from Au–S index data measured at different times ($t$).

$$Au - S\ index = a + \frac{b - a}{1 + \left(\frac{c}{t}\right)^d}. \tag{3}$$

**Data availability**. The authors declare that the main data supporting the findings of this study are available within the article and its Supplementary Information files. All the coding DNA sequences of the proteins used in this study have been deposited in the European Nucleotide Archive (ENA), accession numbers LT986714, LT986711, and LT986708. Extra data are available from the corresponding author upon reasonable request.

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

## Acknowledgements

We appreciate the technical support of Giuliano Siligardi, Rohanah Hussain, and Tamas Javorfi from the Diamond Light Source (Didcot, UK). We thank the Diamond Light Source for experimental beamtime on B23 and the use of the beamline's Zetasizer ZSP (Malvern) for SM12555, SM15232, and SM16042 awarded beamtimes. This work used EPCC's Cirrus HPC Service (https://www.epcc.ed.ac.uk/cirrus). D.A.-M. and E.F. acknowledge support from the Tullow Group Scholarships Programme and Tom West Analytical Fellowship, respectively. The work was in part supported by the European Union's Horizon 2020 research and innovation programme (grant agreement No. 645684 "Immuno-NanoDecoder").

## Author contributions

W.M. performed most of the experiments, analyzed most data and contributed to conceiving the study; A.S. cloned GST-SpyCatcher, purified some of the protein used and performed SRCD experiments; D.R. performed all MD simulations and contributed to manuscript writing; D.A.-M. performed GST binding experiments at different pH and analyzed the data; M.A. performed Raman experiments and data analysis; M.J. cloned GST-SpyTag; F.D.N. performed z-potential measurements; M.B. contributed to Raman data analysis and experimental design; M.S. performed SRCD experiments and contributed to manuscript writing; E.F. contributed to experiments and data analysis, performed SRCD experiments, conceived the study and wrote the manuscript.

## Additional information

**Competing interests:** The authors declare no competing interests.

