## [Peer Review File · Nature Communications]

Reviewers' Comments:

Reviewer #1:

Remarks to the Author:

The manuscript "Modular assembly of proteins on nanoparticles" describes the serial binding of proteins to gold nanoparticles in order to allow a functionalization with a targeting antibody or protein and to prevent the binding of an undefined protein corona. The authors claim that the combination of glutathione S-transferase (GST) with Spy-Catcher plus the model protein engineered to contain the 13-AA long SpyTag provides a platform for stably functionalizing nanoparticles in general (p.3 l.53). They suggest that the functionalization with the SpyTag-SpyCatcher couple and GST leads to a combination of covalent or quasi covalent binding preventing denaturation and loss of functionality of model protein and a protein corona in blood. While the system is very interesting the authors

- 1) did not show that the protein corona is not formed on the final construct. They claim that the binding GST-Au is highly stable and therefore no replacement is expected. As example they use albumin which is one of the first molecules to bind in the corona. Albumin is not the protein/peptide in plasma with the highest affinity for Au NPs but will be replaced after initial binding due to abundance by molecules with a higher affinity to form the "hard" corona. So to confirm their assumption experiments to induce the exchange with peptides or proteins rich in cystin are required.
- 2) fail to explain or show that and why their system is better than avidin-streptavidin. The recombinant inclusion of the SpyTag requires a recombinant production of the model protein which usually is challenging as aggregation and denaturation of the proteins can occur.
- 3) have not shown the general applicability of their system to other nanoparticles. The Au-S bond formed by the GST is highly specific for Au and covalent only on this material. Nanoparticles made from carbon or iron oxide will not show the same binding constant to GST. However, experimental proof of this claim is missing.
- 4) did not consider that the last layer formed by the model protein provides a platform for the binding of a "soft" protein corona, even if the inner "layers" of proteins (GST-SpyCatcher-SpyTag) cannot be replaced.

Reviewer #2:

Remarks to the Author:

This work investigated the binding mechanism of Glutathione S-Transferase (GST) onto gold nanoparticles (GNPs), and confirmed the hypothesis that GST undergoes three steps to bind to GNPs through experiments and modeling. The authors also designed a GST-SpyCatcher fusion protein for modular modification of GNPs. The manuscript is well written and the results nicely presented. However, the SpyTag/SpyCatcher based protein immobilization and material functionalization have been well documented in the past few years. The progress presented in this study is quite specialized and does not represent a significant advancement in the field (Alves et al., 2015; Brune et al., 2016; Giessen and Silver, 2016; Moon et al., 2016; Pessino et al., 2017;

Proschel et al., 2015; Reddington and Howarth, 2015; Stranges et al., 2016). So the reviewer believe that this work should be published in a specialized journal instead of Nature Communications.

Major Comments:

1. Although the Spy chemistry is quite versatile, the GST fusion tag seems to be limited to the modifications of GNPs or gold films because of the presence of accessible Cys residues of GST and the strength of Au-S bond. Then how would "the methods and the principle be adapted to materials other than gold" as claimed in line 53 of main text? Please elaborate.
2. What is the probability difference (or energy difference) at the physisorption stage between cysteine-gold interaction and the salt-bridge conformation based on the modeling results?
3. Instead of using a GST fusion protein, would it be much simpler to genetically introduce two extra cysteines into SpyCatcher for the functionalization of GNPs (Timothy K. Lu et al, Nature Materials 2014)?

Minor Comments:

1. Line 65, Page 4: Several other reports on the use of Spy chemistry for material science should also be cited.
 - i) Engineering Protein Hydrogels Using SpyCatcher-SpyTag Chemistry. Gao X, Fang J, Xue B, Fu L, Li H. *Biomacromolecules*. 2016 Sep 12; 17(9):2812-9.
 - ii) Secrets of a covalent interaction for biomaterials and biotechnology: SpyTag and SpyCatcher. Reddington SC, Howarth M. *Curr Opin Chem Biol*. 2015 Dec; 29:94-9. doi: 10.1016/j.cbpa.2015.10.002. Epub 2015 Oct 30. Review
2. References are needed to justify the assumption that Δd is proportional to the amount of protein that binds to the particles.
3. The unit of protein concentration in Figure 1B should be " μM " instead of " nM ".
4. PDB ID for *S. japonicum* GST should be attached where crystal structure is used.
5. The loading amount in lane1 and lane2 of Figure 4E should be the same, and thus their density should be similar.

References:

- Alves, N.J., Turner, K.B., Daniele, M.A., Oh, E., Medintz, I.L., and Walper, S.A. (2015). Bacterial Nanobioreactors-Directing Enzyme Packaging into Bacterial Outer Membrane Vesicles. *ACS Applied Materials & Interfaces* 7, 24963-24972.
- Brune, K.D., Leneghan, D.B., Brian, I.J., Ishizuka, A.S., Bachmann, M.F., Draper, S.J., Biswas, S., and Howarth, M. (2016). Plug-and-Display: decoration of Virus-Like Particles via isopeptide bonds for modular immunization. *Sci Rep-Uk* 6.
- Giessen, T.W., and Silver, P.A. (2016). A Catalytic Nanoreactor Based on in Vivo Encapsulation of Multiple Enzymes in an Engineered Protein Nanocompartment. *Chembiochem* 17, 1931-1935.
- Moon, H., Bae, Y., Kim, H., and Kang, S. (2016). Plug-and-playable fluorescent cell imaging modular toolkits using the bacterial superglue, SpyTag/SpyCatcher. *Chem Commun (Camb)* 52, 14051-14054.
- Pessino, V., Citron, Y.R., Feng, S., and Huang, B. (2017). Covalent protein labeling by SpyTag-SpyCatcher in fixed cells for super-resolution microscopy. *Chembiochem*.
- Proschel, M., Detsch, R., Boccaccini, A.R., and Sonnewald, U. (2015). Engineering of Metabolic Pathways by Artificial Enzyme Channels. *Front Bioeng Biotechnol* 3, 168.
- Reddington, S.C., and Howarth, M. (2015). Secrets of a covalent interaction for biomaterials and biotechnology: SpyTag and SpyCatcher. *Curr Opin Chem Biol* 29, 94-99.
- Stranges, P.B., Palla, M., Kalachikov, S., Nivala, J., Dorwart, M., Trans, A., Kumar, S., Porel, M.,

Chien, M.C., Tao, C.J., et al. (2016). Design and characterization of a nanopore-coupled polymerase for single-molecule DNA sequencing by synthesis on an electrode array. *P Natl Acad Sci USA* 113, E6749-E6756.

Reviewer #3:

Remarks to the Author:

Review: Modular assembly of proteins on nanoparticles:

Summary: The authors investigated the efficacy of a bio-coupling/bio-conjugation mechanism to assemble proteins on gold nanoparticles. They describe a multi-domain construct; N-terminal of this construct consists of Glutathione S-Transferase (GST) from *Schistosoma japonicum* which has ability to bind gold nanoparticles (GNPs) by forming gold-sulfur bonds (Au-S). The C-terminal of this construct is the SpyCatcher from *Streptococcus pyogenes*, which provides the ability to capture recombinant proteins encoding a SpyTag. The authors investigate the effectiveness of this construct to couple nanoparticles and proteins and show that GST-SpyCatcher activated particles are able to covalently bind a SpyTag expressing protein without losing their functionality.

Major Comments:

1. The main point of the manuscript is to use a special construct to attach nanoparticles to proteins. However, it should be better explained why and how this new construct is better than the conventional methods such as chemisorption via thiol derivatives; (ii) through the use of bi-functional linkers (iii), and through the use of adapter molecules like Streptavidin and biotin. It is not clear why the proposed method is more convenient. No comparison or comments have been made on this.

2. The authors state that: The reported 234 approach has several advantages over other conventional methods: 1. the approach is highly 235 modular, as a SpyTag modified protein would be likely able to bind any GST-SpyCatcher 236 activated surface and vice versa; 2. the method works with materials that are easy to obtain, such 237 as citrate capped GNPs, without the need of tailored surface modifications; 3. the target protein 238 doesn't require chemical modification, but only the recombinant inclusion of a short tag (SpyTag 239 is only 13 amino acids long). First and the last advantages of the proposed method are well established benefits of spy-catcher and SpyTag isopeptide bond and it is well documented in the literature [Reference 1, 2, 3]. These benefits are not the unique part of the methodology described in the manuscript. The second items is the only unique advantage of the proposed methodology but the authors do not explain and detail this aspect.

3. The authors came to a conclusion that the gold nanoparticles-are covalently bonded to SpyCatcher expressing GST since the addition of 2-mercaptethanol (2-ME) did not result any shift/reduction in UV-vis spectra. Although the use of UV-vis spectra is helpful to show the amount of conjugation, it is not convincing to show the strength of the GST-GNP bonding. Additional experimental analyses would be very helpful to complement the author's conclusions such as atomic force microscopy, fluorescence tagging, FRET, optical tweezer etc. Quantifying the bonding strength at different conditions would also strengthen the paper such as at different temperature, pH, different environments/solvents etc. these additional analyses could evidence the importance of the technique.

4. Addition to the previous comment, using control experiments and quantitative comparisons are needed to justify the efficacy of the described methodology and its advantages/benefits over other conventional systems.

5. In figure 2, the authors used 525 and 600nm wavelength to determine Au-S index. It is not clear why 600 nm is chosen and it seems arbitrary. The authors should better explain and justify why this arbitrary wavelength is chosen.

Minor Comments:

1. There is a typo in the manuscript.

We concluded that NEM-GST-SpyCatcher was able to bind to GST-SpyCatcher 223 activated GNPs. (First SpyCatcher should be changed to SpyTag)

2. In Table 1, GST binding properties are summarized. The authors make This is an important result for the manuscript for the paper but Table is not very clear to support the conclusions drawn for these experiments. I suggest to replace the table with the actual measurement plots (supplementary figure1) or co-use this figure with the table.

3. Some important references are missing to cite such as Dr. Mark Howarth's work and Dr. Tirrell's work on SpyTag-SpyCatcher bond. Since the manuscript is built upon the use of this interaction, the papers characterizing the strength and unique properties should be referred.

References

1. Secrets of a covalent interaction for biomaterials and biotechnology: SpyTag and SpyCatcher Samuel C Reddington and Mark Howarth
2. Peptide tag forming a rapid covalent bond to a protein, through engineering a bacterial adhesion by Zakeri et al.
3. Synthesis of bioactive protein hydrogels by genetically encoded SpyTag-SpyCatcher chemistry by Sun et al.

Response to Reviewers' Comments

Reviewer #1 (Remarks to the Author):

The manuscript "Modular assembly of proteins on nanoparticles" describes the serial binding of proteins to gold nanoparticles in order to allow a functionalization with a targeting antibody or protein and to prevent the binding of an undefined protein corona. The authors claim that the combination of glutathione S-transferase (GST) with Spy-Catcher plus the model protein engineered to contain the 13-AA long SpyTag provides a platform for stably functionalizing nanoparticles in general (p.3 l.53). They suggest that the functionalization with the SpyTag-SpyCatcher couple and GST leads to a combination of covalent or quasi covalent binding preventing denaturation and loss of functionality of model protein and a protein corona in blood.

While the system is very interesting the authors

1) did not show that the protein corona is not formed on the final construct. They claim that the binding GST-Au is highly stable and therefore no replacement is expected. As example they use albumin which is one of the first molecules to bind in the corona. Albumin is not the protein/peptide in plasma with the highest affinity for Au NPs but will be replaced after initial binding due to abundance by molecules with a higher affinity to form the "hard" corona. So to confirm their assumption experiments to induce the exchange with peptides or proteins rich in cystin are required.

ANSWER: This is a very important point and we agree that it is essential to show clear evidence that GST prevents the binding of serum proteins to GNPs. To address this, we incubated GST-GNPs and control GNPs in human serum and compared the pattern of bound proteins using SDS-PAGE after a mild wash, as described in the "gel electrophoresis" paragraph in the methods section. We added Figure 1C, Supplementary Figure 1, relevant figure captions and a paragraph to the results section (page 5).

2) fail to explain or show that and why their system is better than avidin-streptavidin. The recombinant inclusion of the SpyTag requires a recombinant production of the model protein which usually is challenging as aggregation and denaturation of the proteins can occur.

ANSWER: The biotin-avidin (or the bacterial analogue streptavidin) system has been overwhelmingly used in prototyping bioconjugation due to the commercial availability of reagents and the simplicity of the method. The practical use of this system is certainly important in molecular diagnostics but less common in the field of biotherapeutics due to limitations such as heterogeneity and poor site-control of the labelling reactions. Also the competition with vitamin H in tissues makes this choice less attractive for in vivo applications. These points have been now introduced and referenced in the conclusive part of the paper. In the same section we also discussed and referenced the increasing number of evidences that recombinant inclusion of SpyTag does not compromise protein structure and function.

3) have not shown the general applicability of their system to other nanoparticles. The Au-S bond formed by the GST is highly specific for Au and covalent only on this material. Nanoparticles made from carbon or iron oxide will not show the same binding constant to GST. However, experimental proof of this claim is missing.

ANSWER: We now included the applicability of this system to silver nanoparticles as well as gold nanoparticles (see Supplementary Figure 6, which essentially replicates the results obtained for gold in Figures 4D and 4E). We expanded the discussion on page 11 on why we believe the stepwise decoration of nanoparticles using SpyCatcher/SpyTag chemistry as mediators could be a general approach, whenever a suitable protein-material interface like GST for gold and silver is available. The set of techniques that we used to characterize GST-gold interface could be used to explore other protein-material pairs, which was beyond the scope of this work but is certainly on our radar. Extra data on silver NPs include SERS and CD data in Figure 2E and Supplementary Figure 4 respectively.

4) did not consider that the last layer formed by the model protein provides a platform for the binding of a "soft" protein corona, even if the inner "layers" of proteins (GST-

SpyCatcher-SpyTag) cannot be replaced.

ANSWER: It is true that the outer layer will eventually have an influence on whether a “soft” corona will form. This will be most likely determined by the exact protein used. In our proof of concept experiment the model protein was GST and we have now shown in Figure 1C that there is no stable association of serum proteins when particles are covered with GST.

Reviewer #2 (Remarks to the Author):

This work investigated the binding mechanism of Glutathione S-Transferase (GST) onto gold nanoparticles (GNPs), and confirmed the hypothesis that GST undergoes three steps to bind to GNPs through experiments and modeling. The authors also designed a GST-SpyCatcher fusion protein for modular modification of GNPs. The manuscript is well written and the results nicely presented. However, the SpyTag/SpyCatcher based protein immobilization and material functionalization have been well documented in the past few years. The progress presented in this study is quite specialized and does not represent a significant advancement in the field (Alves et al., 2015; Brune et al., 2016; Giessen and Silver, 2016; Moon et al., 2016; Pessino et al., 2017; Proschel et al., 2015; Reddington and Howarth, 2015; Stranges et al., 2016). So the reviewer believe that this work should be published in a specialized journal instead of Nature Communications.

Major Comments:

1. Although the Spy chemistry is quite versatile, the GST fusion tag seems to be limited to the modifications of GNPs or gold films because of the presence of accessible Cys residues of GST and the strength of Au-S bond. Then how would “the methods and the principle be adapted to materials other than gold” as claimed in line 53 of main text? Please elaborate.

ANSWER: We expanded this statement in the discussion part on page 11. We replicated the activation of nanoparticles with GST-SpyCatcher and consecutive binding of SpyTag using silver nanoparticles in place of gold and obtained the results shown in Supplementary Figure 6, which are very similar to what has been reported for GNPs in Figure 4. We also argued, in the discussion, that material-protein pairs other than gold-GST could be explored using the same approach that we used in this work, potentially leading to the development of SpyCatcher-activated nanoparticles of different nature than gold and silver.

2. What is the probability difference (or energy difference) at the physisorption stage between cysteine-gold interaction and the salt-bridge conformation based on the modeling results?

ANSWER: This is an interesting point and to address the question we extended the number of MD simulations and confirmed that all the conformations obtained within 20 ns simulation time have salt-bridges with citrate molecules. We revised the relevant text accordingly on page 9 and introduced panels C-E to Supplementary Figure 3. As lysine residues are homogeneously distributed on the surface of the protein, the distance between the cysteine residues and the gold surface varies between simulations. These new results confirm our previous hypothesis of protein physisorption via citrate on gold as a consequence of electrostatic interaction. Compared to the initial, individual simulation, the new dataset suggests that the protein can interact with conformations which orient cysteines towards the gold surface, possibly facilitating chemisorption at larger time scales. It is not easy to obtain absolute values of free energy for such a system using MD, as a much larger sampling would be required to get quantitative thermodynamics. This is beyond the scope of this paper and we used MD primarily to qualitatively assess whether the initial interaction was electrostatically driven. However, an indication of likely values for individual salt-bridge and Au-S interactions have been previously calculated for simpler systems and can be found in the literature. The interaction at physisorption stage between cysteine and gold was estimated around 37.7 kJ/mol in Hoefling et al. 2010. This is qualitatively consistent with a value calculated in our simulation event in which the cysteine was close enough to gold to give a value of interaction energy of ~40 kJ/mol. This value was 9 times smaller than the one observed for the strongest salt-bridge interaction of lysine with citrate in our simulations. The cysteine-gold interactions considered above do not take into account chemisorption and are much smaller than the adsorption energy predicted and measured for Au-S bond (~200 kJ/mol) described for example in Cossaro et al. 2015.

References:

Cossaro, A., Mazzarello, R., Rousseaus, R., Casalis, L., Verdini, A., Kohlmeyer, A., Floreano, L., Scandolo, S., Morgante, A., Klein, M.L., Scoles, G. (2008) X-ray diffraction and computation yield the structure of alkanethiols on gold(111). *Science* **321**:943-946

Hoefling, M., Iori, F., Corni, S., Gottschalk, K.-E. (2010) Interaction of Amino Acids with the Au(111) Surface: Adsorption Free Energies from Molecular Dynamics Simulations. *Langmuir* **26**:8347-8351

3. Instead of using a GST fusion protein, would it be much simpler to genetically introduce two extra cysteines into SpyCatcher for the functionalization of GNPs (Timothy

K. Lu et al, Nature Materials 2014)?

ANSWER: This is a very useful comparison to discuss in this context. Two cysteines at the N-terminal represent an elegant and clearly working alternative to what we have shown here. However, we think that fusion to GST can be achieved equally easily, as GST is an existing affinity tag commonly used in many expression systems. The fusion with SpyCatcher expresses in bacteria well, with high yield and is highly soluble. Importantly, SpyCatcher fused to GST has 3-fold higher binding affinity to gold than SpyCatcher alone. In fact, the values of K_D that we reported for SpyCatcher and GST-SpyCatcher were 1.21 micromolar and 438 nanomolar respectively (see also equilibrium binding curves in Figure 4B). This material has been now included near the end of the manuscript where we also referenced the several relevant papers mentioned in the reviewer's comments.

Minor Comments:

1. Line 65, Page 4: Several other reports on the use of Spy chemistry for material science should also be cited.

i) Engineering Protein Hydrogels Using SpyCatcher-SpyTag Chemistry. Gao X, Fang J, Xue B, Fu L, Li H. *Biomacromolecules*. 2016 Sep 12;17(9):2812-9.

ii) Secrets of a covalent interaction for biomaterials and biotechnology: SpyTag and SpyCatcher. Reddington SC, Howarth M. *Curr Opin Chem Biol*. 2015 Dec;29:94-9. doi: 10.1016/j.cbpa.2015.10.002. Epub 2015 Oct 30. Review

These have been now cited within the text

2. References are needed to justify the assumption that Δd is proportional to the amount of protein that binds to the particles.

Jans et al. 2009 now cited

3. The unit of protein concentration in Figure 1B should be " μM " instead of " nM ".

This has been amended

4. PDB ID for *S. japonicum* GST should be attached where crystal structure is used.

This has been added.

5. The loading amount in lane1 and lane2 of Figure 4E should be the same, and thus their density should be similar.

The experiment has been repeated and the gel shown in the amended figure 4E now shows even loading.

References:

Alves, N.J., Turner, K.B., Daniele, M.A., Oh, E., Medintz, I.L., and Walper, S.A. (2015).

Bacterial Nanobioreactors-Directing Enzyme Packaging into Bacterial Outer Membrane Vesicles. *ACS Applied Materials & Interfaces* 7, 24963-24972.

Brune, K.D., Leneghan, D.B., Brian, I.J., Ishizuka, A.S., Bachmann, M.F., Draper, S.J., Biswas, S., and Howarth, M. (2016). Plug-and-Display: decoration of Virus-Like Particles via isopeptide bonds for modular immunization. *Sci Rep-Uk* 6.

Giessen, T.W., and Silver, P.A. (2016). A Catalytic Nanoreactor Based on in Vivo Encapsulation of Multiple Enzymes in an Engineered Protein Nanocompartment. *Chembiochem* 17, 1931-1935.

Moon, H., Bae, Y., Kim, H., and Kang, S. (2016). Plug-and-playable fluorescent cell imaging modular toolkits using the bacterial superglue, SpyTag/SpyCatcher. *Chem Commun (Camb)* 52, 14051-14054.

Pessino, V., Citron, Y.R., Feng, S., and Huang, B. (2017). Covalent protein labeling by SpyTag-SpyCatcher in fixed cells for super-resolution microscopy. *Chembiochem*.

Proschel, M., Detsch, R., Boccaccini, A.R., and Sonnewald, U. (2015). Engineering of Metabolic Pathways by Artificial Enzyme Channels. *Front Bioeng Biotechnol* 3, 168.

Reddington, S.C., and Howarth, M. (2015). Secrets of a covalent interaction for biomaterials and biotechnology: SpyTag and SpyCatcher. *Curr Opin Chem Biol* 29, 94-99.

Stranges, P.B., Palla, M., Kalachikov, S., Nivala, J., Dorwart, M., Trans, A., Kumar, S., Porel, M., Chien, M.C., Tao, C.J., et al. (2016). Design and characterization of a nanopore-coupled polymerase for single-molecule DNA sequencing by synthesis on an electrode array. *P Natl Acad Sci USA* 113, E6749-E6756.

All these have been now used and referenced in the conclusive part of the paper

Reviewer #3 (Remarks to the Author):

Review: Modular assembly of proteins on nanoparticles:

Summary: The authors investigated the efficacy of a bio-coupling/bio-conjugation mechanism to assemble proteins on gold nanoparticles. They describe a multi-domain construct; N-terminal of this construct consists of Glutathione S-Transferase (GST) from *Schistosoma japonicum* which has ability to bind gold nanoparticles (GNPs) by forming gold-sulfur bonds (Au-S). The C-terminal of this construct is the SpyCatcher from *Streptococcus pyogenes*, which provides the ability to capture recombinant proteins encoding a SpyTag. The authors investigate the effectiveness of this construct to couple nanoparticles and proteins and show that GST-SpyCatcher activated particles are able to covalently bind a SpyTag expressing protein without losing their functionality.

Major Comments

1. The main point of the manuscript is to use a special construct to attach nanoparticles to proteins. However, it should be better explained why and how this new construct is better than the conventional methods such as chemisorption via thiol derivatives; (ii) through the use of bi-functional linkers (iii), and through the use of adapter molecules like Streptavidin and biotin. It is not clear why the proposed method is more convenient. No comparison or comments have been made on this.

ANSWER: We now added comments on page 12 near the end of the manuscript on how our method compares to other conventional and common strategies. We refer mainly to two reviews on the subject (references 3 and 23) and we believe we described now more precisely what the specific advantages of our method are.

2. The authors state that: The reported 234 approach has several advantages over other conventional methods: 1. the approach is highly 235 modular, as a SpyTag modified protein would be likely able to bind any GST-SpyCatcher 236 activated surface and vice versa; 2. the method works with materials that are easy to obtain, such 237 as citrate capped GNPs, without the need of tailored surface modifications; 3. the target protein 238 doesn't require chemical modification, but only the recombinant inclusion of a short tag (SpyTag 239 is only 13 amino acids long).

First and the last advantages of the proposed method are well established benefits of spy-catcher and SpyTag isopeptide bond and it is well documented in the literature [Reference 1, 2, 3]. These benefits are not the unique part of the methodology described in the manuscript. The second items is the only unique advantage of the proposed methodology but the authors do not explain and detail this aspect.

ANSWER: We replaced this text at the end of the manuscript with the comparison mentioned above and we now give more details on the reason why the use of citrate capped NPs would be an advantage. Our method does indeed rely in part on existing technologies such as metal nanoparticles, commercially available recombinant protein expression system, validated and reliable protein-protein interaction systems. However the unique way in which these are combined yields an exceptional one-fits-all modular molecular assembly system suitable for a wide range of applications. The developed method obviates the need for developing unique nanoparticle bio-conjugation methods to match the diversity of protein properties and is therefore suitable for many therapeutic, biotechnology and biomaterials uses and applications.

3. The authors came to a conclusion that the gold nanoparticles-are covalently bonded

to SpyCatcher expressing GST since the addition of 2-mercaptethanol (2-ME) did not result any shift/reduction in UV-vis spectra. Although the use of UV-vis spectra is helpful to show the amount of conjugation, it is not convincing to show the strength of the GST-GNP bonding. Additional experimental analyses would be very helpful to complement the author's conclusions such as atomic force microscopy, fluorescence tagging, FRET, optical tweezer etc. Quantifying the bonding strength at different conditions would also strengthen the paper such as at different temperature, pH, different environments/solvents etc. these additional analyses could evidence the importance of the technique.

ANSWER: We agree that single-molecule techniques would be potentially interesting to characterise this system, however single molecule investigations are beyond the scope of the paper and would better fit a dedicated stand-alone paper. Nevertheless, we applied Surface Enhanced Raman Spectroscopy to provide an extra evidence of covalent Ag-S bond formation in presence of GST but not NEM-GST (Figure 2E) to further support the less direct 2-ME assay.

4. Addition to the previous comment, using control experiments and quantitative comparisons are needed to justify the efficacy of the described methodology and its advantages/benefits over other conventional systems.

ANSWER: As mentioned above, we substantially revised the discussion and included details on how our method compares to other traditional systems. A comprehensive comparison of conjugation methods applicable to nanoparticles was published relatively recently and is cited in reference 23. This is a 170 page review featuring all the conventional and less conventional approaches, each with some unique advantage or limitation in their applicability. Systematic comparison is therefore well beyond the scope of this paper. We think that the most important (or at least generally applicable) comparison to address in protein adsorption is between physisorption and chemisorption, which has been addressed using GST and NEM-GST (Figure 2). On the other hand, specific, direct and quantitative comparison options between our and other established methods would be limited in our opinion, unless restricted to very similar methods. For example, it may be possible to directly compare, using the same set of methods and parameters, Ni-NTA/His-tag and biotin/streptavidin, as these are both affinity-based, non-covalent systems. Our method is substantially different and for its nature could be possibly compared to intein-based conjugation systems, but these have not been developed for nanoparticle conjugation yet. We think that the point made here by the reviewer is fair, but we would be more keen on working on quantitative comparison of half-life of a conjugate *in vivo* and bio-distribution of particles decorated with different methods, rather than merely basing a choice on the strength of the interaction. In our view, *in-vivo* experiments are beyond the scope of this paper but will be considered in the near future.

5. In figure 2, the authors used 525 and 600nm wavelength to determine Au-S index. It is not clear why 600 nm is chosen and it seems arbitrary. The authors should better explain and justify why this arbitrary wavelength is chosen.

ANSWER: We added a clarification (page 7) to better justify 600 nm over other wavelengths. Wavelength over 600 can be used and our choice to use 600 nm was driven by the abundance of filter-based spectrometers and our intention to facilitate and simplify adaptin the method. Wavelengths longer than 600 nm would yield the same trend (for example calculated using 650 nm instead).

Minor Comments:

1. There is a typo in the manuscript.

We concluded that NEM-GST-SpyCatcher was able to bind to GST-SpyCatcher 223 activated GNPs. (First SpyCatcher should be changed to SpyTag)

This has been amended

2. In Table 1, GST binding properties are summarized. The authors make This is an important result for the manuscript for the paper but Table is not very clear to support the conclusions drawn for these experiments. I suggest to replace the table with the actual measurement plots (supplementary figure1) or co-use this figure with the table.

Table and figure are now together (Figure 1E)

3. Some important references are missing to cite such as Dr. Mark Howarth's work and Dr. Tirrell's work on SpyTag-SpyCatcher bond. Since the manuscript is built upon the use of this interaction, the papers characterizing the strength and unique properties should

be referred.

References

1. Secrets of a covalent interaction for biomaterials and biotechnology: SpyTag and SpyCatcher Samuel C Reddington and Mark Howarth

This has been added

2. Peptide tag forming a rapid covalent bond to a protein, through engineering a bacterial adhesion by Zakeri et al.

This was already present

3. Synthesis of bioactive protein hydrogels by genetically encoded SpyTag-SpyCatcher chemistry by Sun et al.

This was already present

Reviewers' Comments:

Reviewer #1:

Remarks to the Author:

The revised version of the manuscript "Modular assembly of proteins on nanoparticles" addressed my critics and comments to the fullest. I am still concerned that the binding strength of gold to thiol is uniquely strong and that other ligands or nanoparticle materials will be modified in the complex matrix of the blood and while passing through the barriers releasing the complex coating. However, this will require for detailed in vivo experiments and will be beyond the scope of this proof-of-concept study.

The other concern I have about the novelty is that the concept to use two high affinity molecules for nanoparticle coating and easy modification is not novel as the authors mentioned several examples of existing methods and discussed their disadvantages. However, the concept of these particular couple of high-affinity binding is quite novel. The procedures are provided in sufficient detail to allow the reproduction by other researchers.

Reviewer #2:

Remarks to the Author:

Despite the incremental improvements, the progress presented in this study does not represent a significant advancement in the field and therefore does not warrant publication in Nature Communications. The referee has provided some relevant references in the first round of reviewing, showing that conclusions are not original. A good number of similar and more sophisticated studies have been published. Shown below are just a few selected examples:

1. SpyCatcher-SpyTag mediated in situ labelling of progeny baculovirus with quantum dots for tracking viral infection in living cells.

Ke X, Zhang Y, Zheng F, Liu Y, Zheng Z, Xu Y, Wang H.

Chem Commun (Camb). 2018 Jan 31;54(10):1189-1192. doi: 10.1039/c7cc08880a.

2. Design and characterization of a nanopore-coupled polymerase for single-molecule DNA sequencing by synthesis on an electrode array.

Stranges PB, Palla M, Kalachikov S, Nivala J, Dorwart M, Trans A, Kumar S, Porel M, Chien M, Tao C, Morozova I, Li Z, Shi S, Aberra A, Arnold C, Yang A, Aguirre A, Harada ET, Korenblum D, Pollard J, Bhat A, Gremyachinskiy D, Bibillo A, Chen R, Davis R, Russo JJ, Fuller CW, Roevers S, Ju J, Church GM.

Proc Natl Acad Sci U S A. 2016 Nov 1;113(44):E6749-E6756. Epub 2016 Oct 11.

3. Programmable polyproteins built using twin peptide superglues.

Veggiani G, Nakamura T, Brenner MD, Gayet RV, Yan J, Robinson CV, Howarth M.

Proc Natl Acad Sci U S A. 2016 Feb 2; 113(5):1202-7. doi: 10.1073/pnas.1519214113. Epub 2016 Jan 19.

Reviewer #3:

Remarks to the Author:

The authors answered all the reviewer comments and made major changes on the discussion of the manuscript. Although additional control experiments are highly recommended as mentioned in the previous review, updated manuscript reads significantly better than the original submission.

Response to Reviewers' Comments

Reviewer #1 (Remarks to the Author):

The revised version of the manuscript "Modular assembly of proteins on nanoparticles" addressed my critics and comments to the fullest. I am still concerned that the binding strength of gold to thiol is uniquely strong and that other ligands or nanoparticle materials will be modified in the complex matrix of the blood and while passing through the barriers releasing the complex coating. However, this will require for detailed *in vivo* experiments and will be beyond the scope of this proof-of-concept study.

The other concern I have about the novelty is that the concept to use two high affinity molecules for nanoparticle coating and easy modification is not novel as the authors mentioned several examples of existing methods and discussed their disadvantages. However, the concept of these particular couple of high-affinity binding is quite novel. The procedures are provided in sufficient detail to allow the reproduction by other researchers.

We agree with the reviewer that *in vivo* experiments would provide an ultimate test for assessing stability of the composite protein-nanomaterial to withstand the *in vivo* environment. Such research is justified but is beyond the scope of our proof-of-concept study. The key principles used and described in our manuscript extend far beyond traditional affinity immobilization approaches. Silver and gold, which were the focus of the study, allow the GST-SpyCatcher fusion to form covalent bonds with the nanoparticle. The attachment of a protein of interest to the GST-derivatised nanoparticles is also covalent and spontaneous. Unlike many variations of methods to covalently link proteins to derivatised nanoparticles by utilising chemically reactive groups, which are often unstable in the presence of water (such as N-hydroxysuccinimide), all of the individual components reported here are generally unreactive. Yet covalent and oriented immobilisation is achieved simply by mixing of proteins and non-derivatized silver or gold nanoparticles under native non-denaturing conditions. We believe that our system is broadly applicable and novel.

With regards to the interface between GST-SpyCatcher and materials other than gold and silver, it is true that the ligands described in the manuscript would require further tests *in vivo* to validate their use in the context of nanomedicine. Irrespective of that, the hierarchical immobilization system described here could be applied in contexts other than *in vivo*, for example in diagnostic kits, biosensors or biocomposite materials to name just a few. Such

and similar applications will benefit from a modular immobilization method, for example in scenarios where multiple proteins need to be linked to multiple materials.

Reviewer #2 (Remarks to the Author):

Despite the incremental improvements, the progress presented in this study does not represent a significant advancement in the field and therefore does not warrant publication in Nature Communications. The referee has provided some relevant references in the first round of reviewing, showing that conclusions are not original. A good number of similar and more sophisticated studies have been published. Shown below are just a few selected examples:

1. SpyCatcher-SpyTag mediated in situ labelling of progeny baculovirus with quantum dots for tracking viral infection in living cells.

Ke X, Zhang Y, Zheng F, Liu Y, Zheng Z, Xu Y, Wang H.

Chem Commun (Camb). 2018 Jan 31;54(10):1189-1192. doi: 10.1039/c7cc08880a.

2. Design and characterization of a nanopore-coupled polymerase for single-molecule DNA sequencing by synthesis on an electrode array.

Stranges PB, Palla M, Kalachikov S, Nivala J, Dorwart M, Trans A, Kumar S, Porel M, Chien M, Tao C, Morozova I, Li Z, Shi S, Aberra A, Arnold C, Yang A, Aguirre A, Harada ET, Korenblum D, Pollard J, Bhat A, Gremyachinskiy D, Bibillo A, Chen R, Davis R, Russo JJ, Fuller CW, Roever S, Ju J, Church GM.

Proc Natl Acad Sci U S A. 2016 Nov 1;113(44):E6749-E6756. Epub 2016 Oct 11.

3. Programmable polyproteins built using twin peptide superglues.

Veggiani G, Nakamura T, Brenner MD, Gayet RV, Yan J, Robinson CV, Howarth M.

Proc Natl Acad Sci U S A. 2016 Feb 2;113(5):1202-7. doi: 10.1073/pnas.1519214113. Epub 2016 Jan 19.

We now included the work from Zhang et al. in Chem Commun to emphasise its relevance for this study. Zhang et al. make use of the same approach as described in reference 57, "Synthesis and patterning of tunable multiscale materials with engineered cells" Nat. Mater. 13, 515–523 (2014), and our discussion now covers both papers. The fact that many recent excellent papers are making use of SpyCatcher/SpyTag is encouraging for the future of the

modular method for the decoration of gold (and other) nanoparticles reported in this manuscript. . We would like to emphasise that the focus of this work is not entirely on the SpyCatcher/SpyTag pair. Combining GST and SpyCatcher yields a unique molecule capable of forming two distinct (soft and hard matter) interfaces. Here we described the design of the key proteins and provided quantitative data regarding GST-gold interface.

Such method for the decoration of gold (and other) nanoparticles that makes use of SpyCatcher/SpyTag could be important for a wider research community, and therefore deserves the attention of the broad readership of Nature Communications.

Reviewer #3 (Remarks to the Author):

The authors answered all the reviewer comments and made major changes on the discussion of the manuscript. Although additional control experiments are highly recommended as mentioned in the previous review, updated manuscript reads significantly better than the original submission.

We are grateful to the reviewer for suggesting in the last revision to expand the comparison between the conjugation method based on GST-SpyCatcher fusion and other existing methods. We included such a comparison in the discussion to improve the manuscript. We understand that the reviewer would also appreciate an experimental comparison, possibly using a single-molecule approach. Although it's not entirely specified which other systems would be appropriate for a direct comparison (streptavidin-biotin was the only one explicitly mentioned), we think that such comparison could be indeed informative, but would be more appropriate if done within a separate publication, possibly in a more specialist journal.